# Morphological and mechanical characterization of bone phenotypes in the Amish G610C murine model of osteogenesis imperfecta

Rachel Kohler, Carli A. Tastad, Amy Creecy, Joseph M. Wallace*

Department of Biomedical Engineering, Indiana University Purdue University of Indianapolis, Indianapolis, IN, United States of America

* jmwalla@iupui.edu

**Data Availability Statement:** The minimal dataset are within the manuscript and its Supporting Information files. Raw dataset can be provided

## Abstract

Osteogenesis imperfecta (OI) is a hereditary bone disease where gene mutations affect Type I collagen formation resulting in osteopenia and increased fracture risk. There are several established mouse models of OI, but some are severe and result in spontaneous fractures or early animal death. The Amish *Col1a2*$^{G610C/+}$ (G610C) mouse model is a newer, moderate OI model that is currently being used in a variety of intervention studies, with differing background strains, sexes, ages, and bone endpoints. This study is a comprehensive mechanical and architectural characterization of bone in G610C mice bred on a C57BL/6 inbred strain and will provide a baseline for future treatment studies. Male and female wild-type (WT) and G610C mice were euthanized at 10 and 16 weeks (n = 13–16). Harvested tibiae, femora, and L4 vertebrae were scanned via micro-computed tomography and analyzed for cortical and trabecular architectural properties. Femora and tibiae were then mechanically tested to failure. G610C mice had less bone but more highly mineralized cortical and trabecular tissue than their sex- and age-matched WT counterparts, with cortical cross-sectional area, thickness, and mineral density, and trabecular bone volume, mineral density, spacing, and number all differing significantly as a function of genotype (2 Way ANOVA with main effects of sex and genotype at each age). In addition, mechanical yield force, ultimate force, displacement, strain, and toughness were all significantly lower in G610C vs. WT, highlighting a brittle phenotype. This characterization demonstrates that despite being a moderate OI model, the Amish G610C mouse model maintains a distinctly brittle phenotype and is well-suited for use in future intervention studies.

## 1. Introduction

Osteogenesis imperfecta (OI) is a hereditary bone disease in which gene mutations affect the formation of Type 1 collagen, leading to weak and brittle bones which can cause severe skeletal deformity and increased fracture risk. OI presentation can vary in severity and is traditionally classified as: Type I, non-deforming (most common); Type II, perinatally lethal; Type III,

upon request to authors. Please contact Joseph Wallace at jmwalla@iupui.edu.

**Funding:** National Institutes of Health (nih.gov), JMW: AR072609 and AC: AR065971. The sponsor played no role in the study design, data collection and analysis, decision to publish, or preparation of the manuscript.

**Competing interests:** The authors have declared that no competing interests exist.

progressively deforming; and Type IV, moderate [1, 2]. This classification scheme, which was initially defined based on clinical presentation, has continued to expand as underlying causes and presentations of OI have been discovered, resulting in 29 heterogeneously distinct forms of OI [3–5]. However, in approximately 90% of cases, OI is caused by defects in COL1A1 or COL1A2 alleles that encode for the Type I collagen alpha 1 and 2 chains, usually in the form of a glycine substitution that inhibits the chains from properly folding into a heterotrimer [4]. The production of mutant polypeptide chains leads to collagen-deficient and defective bone tissue, which creates smaller, fracture-prone bones. One of the most commonly used mouse models of OI, Osteogenesis Imperfecta Murine (oim), is the only naturally occurring murine OI model and features a glycine substitution in the COL1A2 allele resulting in type 1 collagen alpha1 homotrimers [6, 7]. While this substitution genotypically models Ehlers-Danlos syndrome in humans [8, 9], in mice it is phenotypically a brittle bone disease and is used to model OI.

In 2010, an OI model was developed which also has a glycine substitution in the COL1A2 allele, replicating a gene variant found in the Old Order Amish kindred of Lancaster County, PA [6]. While maintaining a small, brittle phenotype [6], the $Col1a2^{G610C/+}$ model is less severe than the oim model, with mice sustaining fewer spontaneous fractures and capable of performing the same exercise regimens as WT mice [10]. These characteristics make the $Col1a2^{G610C/+}$ (G610C) model attractive for intervention studies and biomechanical analysis. Accordingly, the model has been used to investigate the role of mutant collagen in causing OI symptoms such as growth deficiencies [11], defective mineralization of developing bone [12], disrupted osteoblast differentiation [13], and impaired fracture healing [14]. It has also been used in pre-clinical trials of OI treatments such as upregulating the LRP5 pathway [15, 16], diet-based attempts to degrade mutant procollagen and thereby increase bone strength [17, 18], sclerostin antibody and zoledronate combination therapy [19], bone marrow transplant [20], TGF-beta inhibition [16, 21], activin receptor inhibitor treatment to improve muscle contractility [22], myostatin inhibition [23], and BMP-2 injections for fracture healing [24].

While some initial characterization work was performed by Daley *et al.* [6], most studies since have reported only basic geometric and whole bone mechanical properties and have not explored the differences between developmental ages and sex. Previous studies with G610C mice bred on a C57BL/6 background have performed mechanical characterization with 4-pt bending of tibiae [18–20], 3 and 4-pt bending of femora [16–18, 21, 23], compression of vertebrae [18, 19], and tibial torsion [25]. However, these studies are limited in what data are reported and how sex is considered (most examine male or female mice, rarely both). They typically do not report more than basic structural mechanics–most commonly ultimate load, stiffness, and energy to ultimate load [16–20, 23]. Since the incorporation of mutant collagen in the bone matrix is the primary cause of OI's brittle phenotype, tissue-level mechanical properties (e.g. stress, strain, elastic modulus, and toughness where normalization for bone size differences have been performed) are of critical importance to understanding changes in tissue quality. However, only Bi *et al.* have reported tissue-level mechanical properties, as part of their analysis of femora from female mice treated with TGF-β inhibitors [21]. Therefore, to create a baseline for future research and to bolster the mechanical data available in the literature, the current study presents a detailed characterization of bone structure and mechanical behavior of male and female G610C mice bred on a B6 background at crucial developmental ages (10 weeks and 16 weeks).

## 2. Methods

### 2.1 Animals and treatment

All protocols and procedures were performed with prior approval from the Indiana University–Purdue University Indianapolis School of Science Institutional Animal Care and Use

Committee (Protocol #SC296R). Mice used in this study were bred in house and maintained on a C57BL/6J background strain (Jackson Laboratory, Bar Harbor, ME). Male *Col1a2*<sup>G610C/+</sup> (G610C) were housed with pairs of wildtype (WT) females. To genotype WT and G610C off-spring, DNA was extracted from ear notches by first incubating each tissue sample in 50 uL of lysis solution (25 mM NaOH/0.2 mM EDTA) at 90˚C for 1 hour, then neutralizing with 50 uL of neutralization buffer (40 mM Tris HCl). Target genes were amplified with PCR using primers for the G610C forward gene (5′-TCC CTG CTT GCC CTA GTC CCA AAG ATC CTT-3′) and reverse gene (5′-AAG GTA TAG ATC AGA CAG CTG GCA CAT CCA-3′), then run on a 10% agarose gel and imaged with a G8100 E-Gel™ Power Snap Electrophoresis Device (Invitrogen, Carlsbad CA). Mice from each sex and genotype (n = 15-16/group) were euthanized at 10 and 16 weeks of age. Femora, tibiae, and the fourth lumbar spine vertebrae (L4) were harvested, stripped of soft tissue, and frozen in saline-soaked gauze at -20˚C until needed. Some bones were accidentally fractured during harvesting, so testing groups varied by bone (n = 13–15), as detailed in each table.

## 2.2 Microcomputed tomography (μCT) & architectural analysis

To analyze cortical and trabecular architecture, bones were scanned using an isotropic voxel size of 10 μm (Skyscan 1172, Bruker). The right femur, right tibia, and L4 vertebrae from each mouse were scanned through a 0.5 mm Al filter (V = 60kV, I = 167μA) with a 0.7-degree angle increment and two frames averaged. Images were reconstructed (nRecon) and rotated (Data Viewer) before calibrating to hydroxyapatite-mimicking phantoms (0.25 and 0.75 g/cm3 Ca-HA). A 1 mm trabecular region of interest (ROI) was selected at the distal metaphysis of femora and extended proximally from the most proximal portion of the growth plate, and at the proximal metaphysis of the tibia and extending distally from the most distal portion of the growth plate. Trabecular bone in the L4 vertebrae was examined over the entire length of the bone. Cancellous architecture in each ROI was quantified using CT Analyzer (CTAn). A 0.1 mm cortical ROI was selected at approximately 50% length of the femur or tibia, then analyzed with a custom MATLAB (MathWorks, Inc. Natick, MA) program [26]. For visualization, cross-sectional renderings of representative tibia, femora, and vertebrae were created with Drishti Volume Exploration tool [27].

## 2.3 Mechanical testing

Each femur was tested to failure in three-point bending (support span at 7.5 mm for 16 week bones; 6.5 mm for 10 week bones), with the posterior surface in tension. Right tibiae were tested to failure in four-point bending (lower span at 9mm; upper span at 3mm), with the medial surface in tension. Bones were loaded at a displacement control rate of 0.025 mm/s while the sample remained hydrated with PBS. Cross-sectional cortical properties at the fracture site were obtained from μCT images as described above. These properties were used to map load-displacement data into stress-strain data using standard engineering equations as previously reported to estimate tissue level properties [28–30].

## 2.4 Statistical analysis

At each age, data were statistically analyzed using Two-Way ANOVA for main effects of sex and genotype. If a significant interaction (sex X genotype) occurred, the ANOVA was followed by a Tukey post-hoc test. Analysis was performed using GraphPad Prism (v.8) with a significance level at $\alpha = 0.05$.

**Table 1. Mouse weight and bone lengths at 10 and 16 weeks, with p-values from 2-way ANOVA.**

| | MOUSE SIZE | Male | | Female | | p-Values | | |
|---|---|---|---|---|---|---|---|---|
| | | WT (n = 15) | G610C (n = 16) | WT (n = 16) | G610C (n = 16) | Sex | Genotype | Sex * Genotype |
| 10 WKS | Body Weight (g) | 26.2 ± 1.3 | 24.2 ± 2.4 | 20.3 ± 1.2 | 19.6 ± 1.4 | <0.0001 | 0.0029 | 0.1072 |
| | RT length (mm) | 18.2 ± 0.32 | 17.7 ± 0.38 | 17.6 ± 0.55 | 17.3 ± 0.53 | <0.0001 | 0.0018 | 0.1596 |
| | RF length (mm) | 15.4 ± 0.36 | 15 ± 0.47 | 14.9 ± 0.54 | 14.7 ± 0.33 | 0.0002 | 0.002 | 0.8413 |
| | | Male | | Female | | p-Values | | |
| | | WT (n = 15,5) | G610C (n = 15,12) | WT (n = 15,8) | G610C (n = 16,12) | Sex | Genotype | Sex * Genotype |
| **16 WKS** | Body Weight (g)* | 26.9 ± 1.2 | 26.1 ± 2.1 | 23.5 ± 1 | 21.7 ± 1.5 | <0.0001 | 0.0252 | 0.4327 |
| | RT length (mm) | 18.3 ± 0.31 | 17.7 ± 0.84 | 18.2 ± 0.43 | 17.9 ± 0.4 | 0.4853 | 0.0009 | 0.6501 |
| | RF Length (mm) | 15.9 ± 0.21 | 15.6 ± 0.36 | 16 ± 0.17 | 15.7 ± 0.43 | 0.4605 | 0.0003 | 0.9032 |

*Body weight data for the first round of sacced mice were lost. (n = a,b) indicates sample size for a = overall, b = body weight.

Data presented as mean +/- standard deviation.

## 3. Results

### 3.1 Body mass and bone length

Table 1 shows that G610C mice and their bones were consistently smaller than WT mice, with main effects of genotype for body weight and long bone length at both ages. In contrast, sex-based differences between bone lengths lessened as the mice reached skeletal maturity, although body weight differences between sexes persisted.

### 3.2 Cortical bone phenotype in femora and tibiae

G610C mice had less, more highly mineralized cortical bone than their sex- and age-matched WT counterparts. With the exception of cortical thickness, there was a main effect of genotype ($p \leq 0.0062$ for all) for every cortical measure in both male and female mice, at both age points, and in both femora and tibiae (Tables 2 and 3). G610C bones were smaller overall as compared to WT bones (Fig 1, S1 and S2 Figs), highlighted by smaller total area, bone area, maximum moment of inertia (Imax), and minimum moment of inertia (Imin) as compared to WT mice. Tissue mineral density (TMD) was higher in G610C bones versus WT. There was also a significant effect of sex ($p \leq 0.0215$) for almost every property investigated, except for 16-week tibial TMD and femoral cortical thickness, as the male mice tended to be larger than the female mice. There were significant interactions in only two measures: 10-week tibial marrow area ($p = 0.0054$) and 10-week femoral marrow area ($p = 0.0466$), with post-hoc analyses showing the same behavior of WT mice having larger marrow area than G610C mice.

### 3.3 Trabecular phenotype

G610C mice had less trabecular bone than their sex- and age-matched WT counterparts. There was a significant main effect ($p \leq 0.0362$) of genotype for nearly every trabecular measure investigated. (Tables 4 and 5). OI mice had sparse, small trabeculae (Fig 2, S3–S5 Figs), as evidenced by lower bone volume fraction (BV/TV), trabecular thickness (Tb.Th), and trabecular number (Tb.N), and higher trabecular spacing (Tb.Sp), as compared to WT mice. In contrast to the cortical phenotype, the TMD phenotype in trabecular bone flipped between 10 and 16-week mice. At 10 weeks, female G610C had lower TMD than WT mice but by 16 weeks of age, G610C trabecular TMD was higher than WT in both sexes. As with cortical bone, there was a significant ($p \leq 0.0233$) main effect of sex for most trabecular properties, with male mice generally having denser, more mineralized trabecular bone than female mice. There was a

**Table 2. Tibia cortical properties at 10 and 16 weeks, with p-values from 2-way ANOVA.**

| | TIBIA CORTICAL PROPERTIES | Male | | Female | | P-value | | |
|---|---|---|---|---|---|---|---|---|
| | | WT (n = 15) | G610C (n = 15) | WT (n = 14) | G610C (n = 13) | Sex | Genotype | Sex * Genotype |
| 10 WKS | Total Area (mm$^2$) | 1.46 ± 0.12 | 1.25 ± 0.13 | 1.17 ± 0.09 | 1.03 ± 0.07 | <0.0001 | <0.0001 | 0.2504 |
| | Marrow Area (mm$^2$) | 0.61 ± 0.05 | 0.45 ± 0.05 * | 0.49 ± 0.05 $ | 0.41 ± 0.04 *$ | <0.0001 | <0.0001 | **0.0054** |
| | Bone Area (mm$^2$) | 0.85 ± 0.08 | 0.80 ± 0.10 | 0.68 ± 0.05 | 0.62 ± 0.04 | <0.0001 | **0.0036** | 0.8939 |
| | Cortical Thickness (mm) | 0.24 ± 0.01 | 0.24 ± 0.02 | 0.21 ± 0.01 | 0.21 ± 0.01 | <0.0001 | 0.8721 | 0.1537 |
| | Imax (mm$^4$) | 0.25 ± 0.05 | 0.21 ± 0.05 | 0.16 ± 0.03 | 0.13 ± 0.03 | <0.0001 | **0.0007** | 0.3972 |
| | Imin (mm$^4$) | 0.10 ± 0.02 | 0.08 ± 0.02 | 0.06 ± 0.01 | 0.05 ± 0.01 | <0.0001 | <0.0001 | 0.1679 |
| | TMD (g/cm$^3$ HA) | 1.15 ± 0.01 | 1.20 ± 0.03 | 1.18 ± 0.02 | 1.22 ± 0.02 | <0.0001 | <0.0001 | 0.2971 |
| | | Male | | Female | | P-value | | |
| | | WT (n = 15) | G610C (n = 14) | WT (n = 15) | G610C (n = 15) | Sex | Genotype | Sex * Genotype |
| 16 WKS | Total Area (mm$^2$) | 1.38 ± 0.09 | 1.22 ± 0.14 | 1.26 ± 0.06 | 1.05 ± 0.05 | <0.0001 | <0.0001 | 0.3329 |
| | Marrow Area (mm$^2$) | 0.53 ± 0.04 | 0.42 ± 0.05 | 0.48 ± 0.04 | 0.35 ± 0.04 | <0.0001 | <0.0001 | 0.2435 |
| | Bone Area (mm$^2$) | 0.86 ± 0.07 | 0.80 ± 0.10 | 0.78 ± 0.03 | 0.7 ± 0.04 | <0.0001 | **0.0002** | 0.5324 |
| | Cortical Thickness (mm) | 0.25 ± 0.01 | 0.25 ± 0.02 | 0.24 ± 0.01 | 0.24 ± 0.01 | **0.0133** | 0.3249 | 0.9019 |
| | Imax (mm$^4$) | 0.26 ± 0.04 | 0.23 ± 0.06 | 0.21 ± 0.02 | 0.17 ± 0.02 | <0.0001 | **0.0001** | 0.5648 |
| | Imin (mm$^4$) | 0.09 ± 0.01 | 0.08 ± 0.02 | 0.07 ± 0.01 | 0.05 ± 0.01 | <0.0001 | <0.0001 | 0.3742 |
| | TMD (g/cm$^3$ HA) | 1.28 ± 0.15 | 1.38 ± 0.20 | 1.27 ± 0.15 | 1.43 ± 0.17 | 0.6328 | **0.0062** | 0.4712 |

Significant effects (p>0.05) are highlighted in gray. Significant post-hoc values (p<0.05) between genotypes are marked with *, and between sex are marked with $. Data presented as mean +/- standard deviation.

significant interaction for trabecular thickness (10-week tibial, p = 0.0037; 16-week tibial, p = 0.0011; 16-wek femoral, p = 0.0013) and 16-week tibial TMD (p = 0.043). Post-hoc analysis showed that female mice exhibited a greater disparity in trabecular spacing and TMD between

**Table 3. Femur cortical properties at 10 and 16 weeks, with p-values from 2-way ANOVA.**

| | FEMUR CORTICAL PROPERTIES | Male | | Female | | P-value | | |
|---|---|---|---|---|---|---|---|---|
| | | WT (n = 15) | G610C (n = 13) | WT (n = 15) | G610C (n = 13) | Sex | Genotype | Sex * Genotype |
| 10 WKS | Total Area (mm$^2$) | 2.01 ± 0.16 | 1.70 ± 0.17 | 1.69 ± 0.11 | 1.44 ± 0.11 | <0.0001 | <0.0001 | 0.4311 |
| | Marrow Area (mm$^2$) | 1.12 ± 0.08 | 0.87 ± 0.10 * | 0.97 ± 0.08 $ | 0.81 ± 0.09 * | <0.0001 | <0.0001 | **0.0466** |
| | Bone Area (mm$^2$) | 0.89 ± 0.09 | 0.83 ± 0.13 | 0.72 ± 0.06 | 0.62 ± 0.04 | <0.0001 | **0.0009** | 0.4546 |
| | Cortical Thickness (mm) | 0.20 ± 0.01 | 0.21 ± 0.03 | 0.18 ± 0.01 | 0.17 ± 0.01 | <0.0001 | 0.5226 | 0.1292 |
| | Imax (mm$^4$) | 0.31 ± 0.05 | 0.23 ± 0.05 | 0.21 ± 0.03 | 0.15 ± 0.02 | <0.0001 | <0.0001 | 0.2836 |
| | Imin (mm$^4$) | 0.16 ± 0.03 | 0.12 ± 0.03 | 0.11 ± 0.01 | 0.08 ± 0.01 | <0.0001 | <0.0001 | 0.8004 |
| | TMD (g/cm$^3$ HA) | 1.28 ± 0.02 | 1.36 ± 0.02 | 1.31 ± 0.02 | 1.36 ± 0.03 | **0.0215** | <0.0001 | 0.0745 |
| | | Male | | Female | | P-value | | |
| | | WT (n = 15) | G610C (n = 15) | WT (n = 15) | G610C (n = 15) | Sex | Genotype | Sex * Genotype |
| 16 WKS | Total Area (mm$^2$) | 1.88 ± 0.1 | 1.65 ± 0.16 | 1.73 ± 0.09 | 1.43 ± 0.07 | <0.0001 | <0.0001 | 0.1973 |
| | Marrow Area (mm$^2$) | 1.06 ± 0.05 | 0.85 ± 0.08 | 0.94 ± 0.06 | 0.74 ± 0.06 | <0.0001 | <0.0001 | 0.9763 |
| | Bone Area (mm$^2$) | 0.82 ± 0.06 | 0.79 ± 0.10 | 0.79 ± 0.04 | 0.7 ± 0.06 | **0.0008** | **0.0004** | 0.0508 |
| | Cortical Thickness (mm) | 0.19 ± 0.01 | 0.20 ± 0.02 | 0.20 ± 0.01 | 0.19 ± 0.02 | 0.5941 | 0.2812 | 0.0553 |
| | Imax (mm$^4$) | 0.27 ± 0.04 | 0.23 ± 0.04 | 0.22 ± 0.02 | 0.16 ± 0.02 | <0.0001 | <0.0001 | 0.324 |
| | Imin (mm$^4$) | 0.13 ± 0.02 | 0.11 ± 0.03 | 0.13 ± 0.01 | 0.09 ± 0.01 | <0.0001 | **0.002** | 0.1067 |
| | TMD (g/cm$^3$ HA) | 1.37 ± 0.02 | 1.42 ± 0.02 | 1.39 ± 0.02 | 1.46 ± 0.03 | <0.0001 | <0.0001 | 0.6548 |

Significant effects (p>0.05) are highlighted in gray. Significant post-hoc values (p<0.05) between genotypes are marked with *, and between sex are marked with $. Data presented as mean +/- standard deviation.

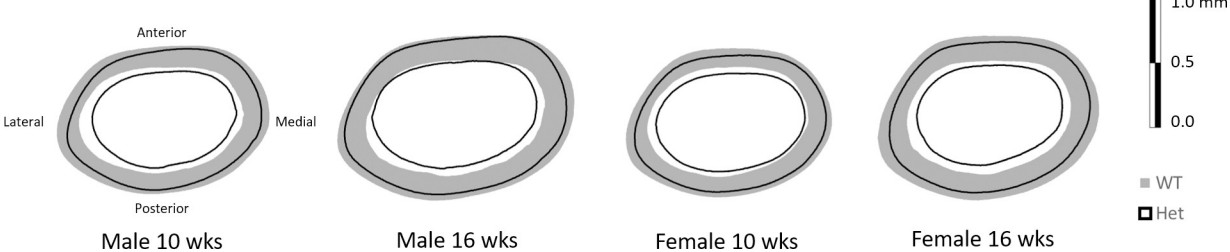

**Fig 1. Schematic transverse cross sections from femoral mid-diaphysis.** Shown are average profiles of the femoral cortical ROIs from all bones in each group, demonstrating that cortical bone area is consistently smaller in G610C mice in both ages and sexes.

genotypes than male mice. Similar results were also seen in trabecular bone of L4 vertebrae (S1 Table).

### 3.4 Mechanical properties

Bending tests demonstrated that the G610C bones exhibit brittle behavior in comparison to WT bones, a phenotype that was consistent in both bones and age groups (Figs 3 and 4). There was a main effect ($p \leq 0.0309$) of genotype for most structural mechanical properties. The smaller G610C bones sustained a lower yield force (only in tibiae), ultimate force, displacement to yield (only in femora), postyield displacement, total displacement, stiffness, and work, as compared to WT bones. However, trends in estimated tissue-level properties differed. There was a significant ($p \leq 0.0263$) main effect of genotype for total strain and toughness in the tibiae, while a main effect of genotype was seen for all properties investigated in the femur except for 10-week strain and work to yield. There were significant ($p \leq 0.0489$) main effects of sex for many mechanical properties as well, including yield force, ultimate force, work to yield, ultimate stress, and strain to yield, with males consistently having higher properties than females (Tables 6 and 7). There was a single interaction effect in 16-week femoral toughness ($p = 0.044$), but there were no specific post-hoc differences in comparisons of interest.

**Table 4. Tibia trabecular properties at 10 and 16 weeks, with p-values from 2-way ANOVA.**

|  | TIBIA TRABECULAR PROPERTIES | Male | | Female | | P-value | | |
|---|---|---|---|---|---|---|---|---|
|  |  | WT (n = 15) | G610C (n = 15) | WT (n = 14) | G610C (n = 13) | Sex | Genotype | Sex * Genotype |
| **10 WKS** | BV/TV (%) | 24.4 ± 5.8 | 20.8 ± 5.6 | 16.1 ± 4.1 | 11.3 ± 1.3 | <0.0001 | 0.0005 | 0.7624 |
|  | Tb.Th (μm) | 60.5 ± 4.1 | 57.9 ± 4.8 | 57.8 ± 2.5 | 52.7 ± 1.7 | 0.0002 | <0.0001 | 0.2817 |
|  | Tb.Sp (mm) | 0.14 ± 0.01 | 0.16 ± 0.02 | 0.20 ± 0.02 $ | 0.25 ± 0.03 *$ | <0.0001 | <0.0001 | 0.0037 |
|  | Tb.N (1/mm) | 3.99 ± 0.71 | 3.55 ± 0.70 | 2.78 ± 0.64 | 2.15 ± 0.22 | <0.0001 | 0.0008 | 0.6852 |
|  | TMD (g/cm³ HA) | 0.69 ± 0.02 | 0.69 ± 0.02 | 0.68 ± 0.02 | 0.65 ± 0.02 *$ | 0.0002 | 0.0001 | 0.043 |
|  |  | Male | | Female | | P-value | | |
|  |  | WT (n = 15) | G610C (n = 14) | WT (n = 15) | G610C (n = 15) | Sex | Genotype | Sex * Genotype |
| **16 WKS** | BV/TV (%) | 20.7 ± 3.1 | 18 ± 4.9 | 14.5 ± 2.5 | 9.3 ± 2.2 | <0.0001 | <0.0001 | 0.1517 |
|  | Tb.Th (μm) | 62.1 ± 4.2 | 61.6 ± 3.8 | 63.4 ± 1.8 | 64.5 ± 4 | 0.0233 | 0.7398 | 0.3844 |
|  | Tb.Sp (mm) | 0.17 ± 0.01 | 0.18 ± 0.02 | 0.23 ± 0.03 $ | 0.28 ± 0.04 *$ | <0.0001 | <0.0001 | 0.0011 |
|  | Tb.N (1/mm) | 3.33 ± 0.37 | 2.91 ± 0.63 | 2.29 ± 0.45 | 1.45 ± 0.35 | <0.0001 | <0.0001 | 0.0841 |
|  | TMD (g/cm³ HA) | 0.71 ± 0.03 | 0.73 ± 0.04 | 0.72 ± 0.02 | 0.75 ± 0.06 | 0.3952 | 0.011 | 0.5492 |

Significant effects (p>0.05) are highlighted in gray. Significant post-hoc values (p<0.05) between genotypes are marked with *, and between sex are marked with $. Data presented as mean +/- standard deviation.

**Table 5. Femur trabecular properties at 10 and 16 weeks, with p-values from 2-way ANOVA.**

| | FEMUR TRABECULAR PROPERTIES | Male | | Female | | P-value | | |
|---|---|---|---|---|---|---|---|---|
| | | WT (n = 15) | G610C (n = 13) | WT (n = 15) | G610C (n = 13) | Sex | Genotype | Sex * Genotype |
| **10 WKS** | BV/TV (%) | 23.4 ± 3.4 | 17.1 ± 4.9 | 12.7 ± 3.3 | 7.4 ± 1.2 | <0.0001 | <0.0001 | 0.4796 |
| | Tb.Th (μm) | 62.4 ± 4.7 | 59.6 ± 5.5 | 58.1 ± 3.5 | 55.6 ± 3.8 | 0.0013 | 0.0362 | 0.8258 |
| | Tb.Sp (mm) | 0.16 ± 0.01 | 0.20 ± 0.04 | 0.21 ± 0.02 | 0.26 ± 0.01 | <0.0001 | <0.0001 | 0.4397 |
| | Tb.N (1/mm) | 3.73 ± 0.33 | 2.84 ± 0.61 | 2.17 ± 0.48 | 1.32 ± 0.17 | <0.0001 | <0.0001 | 0.6646 |
| | TMD (g/cm$^3$ HA) | 0.70 ± 0.02 | 0.70 ± 0.03 | 0.68 ± 0.02 | 0.66 ± 0.02 | <0.0001 | 0.0302 | 0.2691 |
| | | Male | | Female | | P-value | | |
| | | WT (n = 15) | G610C (n = 15) | WT (n = 15) | G610C (n = 15) | Sex | Genotype | Sex * Genotype |
| **16 WKS** | BV/TV (%) | 20.0 ± 3.5 | 15.4 ± 4.4 | 10.6 ± 2.0 | 7.1 ± 0.9 | <0.0001 | <0.0001 | 0.4832 |
| | Tb.Th (μm) | 61.2 ± 4.5 | 59.5 ± 5.5 | 59.5 ± 2.1 | 60.2 ± 3.9 | 0.6633 | 0.6517 | 0.26 |
| | Tb.Sp (mm) | 0.20 ± 0.01 | 0.20 ± 0.01 * | 0.24 ± 0.02 $ | 0.30 ± 0.02 *$ | <0.0001 | <0.0001 | 0.0013 |
| | Tb.N (1/mm) | 3.25 ± 0.4 | 2.56 ± 0.53 | 1.79 ± 0.35 | 1.19 ± 0.18 | <0.0001 | <0.0001 | 0.6244 |
| | TMD (g/cm$^3$ HA) | 0.79 ± 0.095 | 0.76 ± 0.097 | 0.74 ± 0.099 | 0.76 ± 0.101 | 0.4065 | 0.7282 | 0.3623 |

Significant effects (p>0.05) are highlighted in gray. Significant post-hoc values (p<0.05) between genotypes are marked with *, and between sex are marked with $. Data presented as mean +/- standard deviation.

## 4. Discussion

It is well established that fracture risk is a function of both bone size and tissue quality. Intuitively, small bones generally fracture at lower loads than larger bones. A hallmark of OI bone is brittle mechanical behavior, reflected in reduced post-yield deformation and toughness. These properties have less to do with bone size and strength, and more to do with quality and deformability of the tissue. OI is characterized by bones that are not only small, but also suffer from inferior tissue quality due to mutant collagen being incorporated into the bone matrix thereby inhibiting normal cross-linking, mineralization, and hydration [31]. A representative model of this disease should not only have smaller bones with reduced structural mechanical properties, but also tissue-level mechanical inferiorities.

The G610C model of OI is well suited for investigating bone mechanical deficits and potential interventions. Several other OI models exist, but most suffer from limitations that make them less suitable for studies looking at mechanical stimulation (as bones may break in vivo),

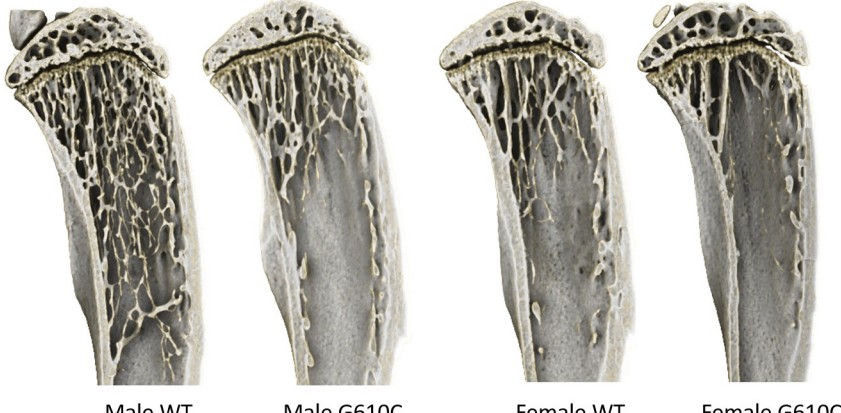

Male WT          Male G610C          Female WT          Female G610C

**Fig 2. Sagittal tibial cross sections.** Shown are cross-sectional views of 10-wk tibia from representative mice (the trend is similar in 16-wk mice). Trabecular bone quantity varies significantly between genotypes.

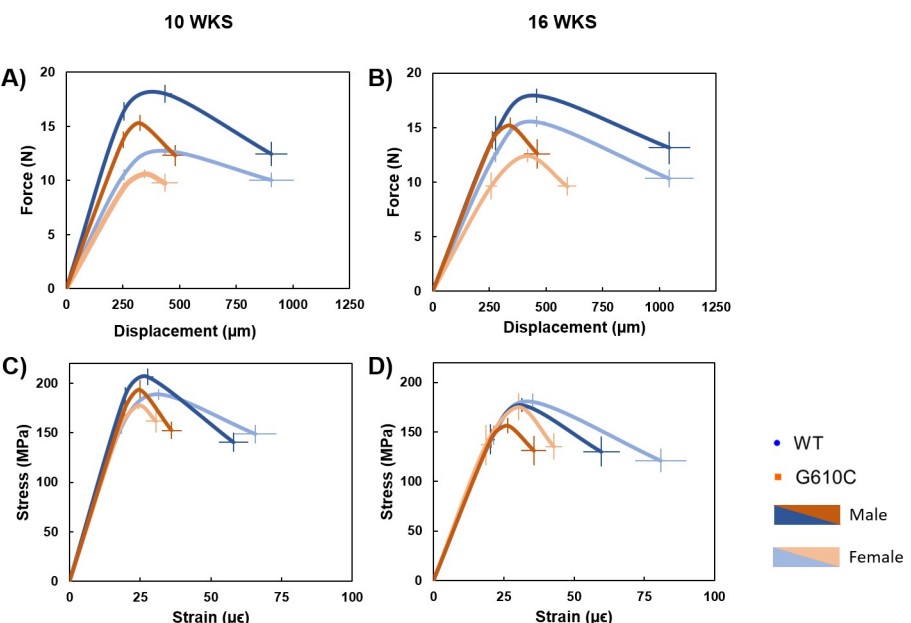

**Fig 3. Schematic mechanical curves from tibial 4-pt bending tests. A)** Average force-displacement plot for 10-week mice, showing weaker G610C bones with significant deficiencies in postyield and total displacement. **B)** Strength and deformation disparities continued with the 16-week bones. **C)** Average stress-strain plots for 10-week mice demonstrates that G610C tissue breaks at lower strains than WT bones. **D)** This brittle phenotype is maintained at 16-weeks of age. Data points are mean values +/- standard error.

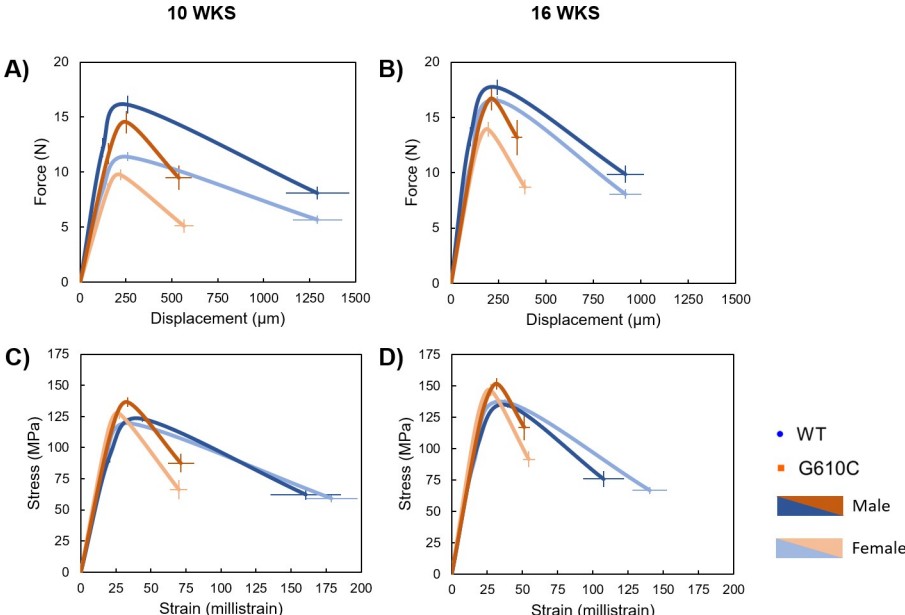

**Fig 4. Schematic mechanical curves from femoral 3-pt bending tests. A)** Average force-displacement plot for 10-week mice, showing weaker G610C bones with significant deficiencies in postyield and total displacement. **B)** Strength and deformation disparities continued with the 16-week bones. **C)** Average stress-strain plots for 10-week mice demonstrate that G610C tissue breaks at lower strains than WT bones. **D)** This brittle phenotype is maintained at 16-weeks of age. Data points are mean values +/- standard error.

**Table 6. Tibia mechanical properties at 10 and 16 weeks, with p-values from 2-way ANOVA.**

| | TIBIAL MECHANICS | Male | | Female | | P-value | | |
|---|---|---|---|---|---|---|---|---|
| | | WT (n = 15) | G610C (n = 15) | WT (n = 14) | G610C (n = 13) | Sex | Genotype | Sex * Genotype |
| **10 WKS** | Yield Force (N) | 16.4 ± 2.8 | 13.8 ± 2.5 | 10.5 ± 1.8 | 9.3 ± 1.6 | <0.0001 | 0.0031 | 0.1995 |
| | Ultimate Force (N) | 18.0 ± 2.7 | 15.3 ± 2.5 | 12.7 ± 1.5 | 10.6 ± 1.5 | <0.0001 | <0.0001 | 0.6247 |
| | Displacement to Yield (μm) | 250 ± 13 | 250 ± 29 | 254 ± 20 | 252 ± 20 | 0.6885 | 0.892 | 0.8758 |
| | Postyield Displacement (μm) | 496 ± 230 | 231 ± 179 | 651 ± 347 | 182 ± 207 | 0.4195 | <0.0001 | 0.1231 |
| | Total Displacement (μm) | 746 ± 236 | 481.6 ± 189 | 904.1 ± 341 | 434 ± 197 | 0.3986 | <0.0001 | 0.1189 |
| | Stiffness (N/mm) | 74 ± 15 | 62 ± 12 | 47 ± 8 | 42 ± 7 | <0.0001 | 0.0054 | 0.256 |
| | Work to Yield (mJ) | 2.20 ± 0.34 | 1.85 ± 0.45 | 1.45 ± 0.28 | 1.26 ± 0.23 | <0.0001 | 0.0037 | 0.3478 |
| | Postyield Work (mJ) | 7.15 ± 2.96 | 2.91 ± 2.16 | 6.94 ± 3.37 | 1.51 ± 1.42 | 0.2411 | <0.0001 | 0.3855 |
| | Total Work (mJ) | 9.35 ± 3.05 | 4.77 ± 2.25 | 8.39 ± 3.32 | 2.78 ± 1.31 | 0.0352 | <0.0001 | 0.4583 |
| | Yield Stress (MPa) | 187.5 ± 28 | 172.9 ± 29.5 | 155.3 ± 21.4 | 155.4 ± 16.8 | 0.0003 | 0.2694 | 0.2579 |
| | Ultimate Stress (MPa) | 206 ± 28 | 193 ± 32 | 189 ± 19 | 178 ± 15 | 0.0118 | 0.0664 | 0.8841 |
| | Strain to Yield (mε) | 19.7 ± 1.2 | 19.4 ± 2.5 | 18.3 ± 1.2 | 17.9 ± 1.7 | 0.0033 | 0.467 | 0.9151 |
| | Total Strain (mε) | 58.1 ± 17.2 | 36.1 ± 11.4 | 65.7 ± 25.8 | 30.6 ± 12.8 | 0.8199 | <0.0001 | 0.161 |
| | Modulus (GPa) | 10.72 ± 1.95 | 10.12 ± 2.26 | 9.54 ± 1.17 | 9.83 ± 1.15 | 0.1096 | 0.7313 | 0.3304 |
| | Resilience (MPa) | 1.98 ± 0.26 | 1.80 ± 0.37 | 1.54 ± 0.26 | 1.51 ± 0.25 | <0.0001 | 0.1562 | 0.345 |
| | Toughness (MPa) | 8.51 ± 2.95 | 4.68 ± 2.26 | 8.97 ± 3.48 | 3.36 ± 1.69 | 0.5491 | <0.0001 | 0.2116 |
| | | Male | | Female | | P-value | | |
| | | WT (n = 15) | G610C (n = 14) | WT (n = 15) | G610C (n = 15) | Sex | Genotype | Sex * Genotype |
| **16 WKS** | Yield Force (N) | 14.5 ± 3.8 | 13.7 ± 2.6 | 12.6 ± 2.6 | 9.68 ± 3.7 | 0.0009 | 0.0309 | 0.2093 |
| | Ultimate Force (N) | 18.0 ± 2.2 | 15.3 ± 2.5 | 15.5 ± 1.8 | 12.4 ± 2.2 | <0.0001 | <0.0001 | 0.6895 |
| | Displacement to Yield (μm) | 243 ± 51 | 256 ± 22 | 260 ± 33 | 245 ± 64 | 0.8002 | 0.9492 | 0.248 |
| | Postyield Displacement (μm) | 504 ± 277 | 226 ± 202 | 767 ± 362 | 345 ± 184 | 0.008 | <0.0001 | 0.3053 |
| | Total Displacement (μm) | 747.1 ± 262 | 482.4 ± 201 | 1027.1 ± 355 | 590.7 ± 145 | 0.0049 | <0.0001 | 0.2001 |
| | Stiffness (N/mm) | 68 ± 11 | 60 ± 13 | 54 ± 11 | 44 ± 10 | <0.0001 | 0.0042 | 0.6681 |
| | Work to Yield (mJ) | 1.94 ± 0.65 | 1.86 ± 0.38 | 1.77 ± 0.50 | 1.35 ± 0.64 | 0.0214 | 0.0961 | 0.2419 |
| | Postyield Work (mJ) | 6.99 ± 2.73 | 2.89 ± 2.67 | 9.29 ± 4.68 | 3.41 ± 1.44 | 0.0881 | <0.0001 | 0.2808 |
| | Total Work (mJ) | 8.93 ± 2.4 | 4.75 ± 2.65 | 11.05 ± 4.79 | 4.76 ± 1.08 | 0.1843 | <0.0001 | 0.1887 |
| | Yield Stress (MPa) | 142.7 ± 35.5 | 141.6 ± 28.5 | 146 ± 31.5 | 137.3 ± 61.9 | 0.9665 | 0.6518 | 0.7285 |
| | Ultimate Stress (MPa) | 177 ± 24 | 158 ± 26 | 180 ± 28 | 175 ± 46 | 0.2269 | 0.1474 | 0.4158 |
| | Strain to Yield (mε) | 19.3 ± 4.4 | 19.9 ± 1.9 | 20.0 ± 2.6 | 17.8 ± 4.7 | 0.4376 | 0.3919 | 0.144 |
| | Total Strain (mε) | 58.8 ± 18.8 | 37.4 ± 15.5 | 79.5 ± 29.4 | 42.6 ± 10.2 | 0.0151 | <0.0001 | 0.1403 |
| | Modulus (GPa) | 8.33 ± 1.05 | 7.92 ± 1.50 | 8.18 ± 1.73 | 8.71 ± 3.10 | 0.542 | 0.9102 | 0.3687 |
| | Resilience (MPa) | 1.52 ± 0.52 | 1.51 ± 0.36 | 1.57 ± 0.44 | 1.38 ± 0.69 | 0.7729 | 0.4471 | 0.5229 |
| | Toughness (MPa) | 7.05 ± 2.32 | 3.77 ± 2.00 | 9.64 ± 3.76 | 4.78 ± 1.15 | 0.008 | <0.0001 | 0.2327 |

Significant effects (p>0.05) are highlighted in gray. Significant post-hoc values (p<0.05) between genotypes are marked with *, and between sex are marked with $. Data presented as mean +/- standard deviation.

or where mechanical endpoints are planned (because of sample loss due to spontaneous fracture and the need for large sample sizes to overcome variability in mechanical analyses). The popular (and commercially available) homozygous oim model exhibits a severe phenotype, leading to mice often sustaining spontaneous long bone fractures during life and an increased risk of specimen loss during tissue harvesting and handling [32]. While mechanical properties are markedly deficient, even more so than G610C vs. WT, this combination of poor health and brittleness makes it difficult to properly power loading intervention studies. The Brtl IV model, another common OI model, has a moderate bone phenotype and mechanical analyses

**Table 7. Femur mechanical properties at 10 and 16 weeks, with p-values from 2-way ANOVA.**

| | FEMORAL MECHANICS | Male | | Female | | P-value | | |
|---|---|---|---|---|---|---|---|---|
| | | WT (n = 15) | G610C (n = 13) | WT (n = 15) | G610C (n = 13) | Sex | Genotype | Sex * Genotype |
| **10 WKS** | Yield Force (N) | 12.2 ± 2.8 | 11.66 ± 3.2 | 8.4 ± 1.9 | 8.5 ± 1.6 | <0.0001 | 0.7078 | 0.6313 |
| | Ultimate Force (N) | 16.1 ± 2.8 | 14.6 ± 3.4 | 11.4 ± 1.5 | 9.8 ± 1.7 | <0.0001 | 0.0166 | 0.9648 |
| | Displacement to Yield (µm) | 146 ± 33 | 155 ± 25 | 125 ± 20 | 152 ± 20 | 0.0775 | 0.0093 | 0.1812 |
| | Postyield Displacement (µm) | 975 ± 554 | 382 ± 251 | 1168 ± 458 | 414 ± 186 | 0.3008 | <0.0001 | 0.4541 |
| | Total Displacement (µm) | 1121 ± 568 | 537.5 ± 239 | 1293.4 ± 465 | 565.8 ± 183 | 0.3623 | <0.0001 | 0.5123 |
| | Stiffness (N/mm) | 95 ± 21 | 84 ± 24 | 77 ± 10 | 63 ± 9 | <0.0001 | 0.0115 | 0.7905 |
| | Work to Yield (mJ) | 1.01 ± 0.39 | 1.00 ± 0.332 | 0.61 ± 0.20 | 0.71 ± 0.20 | <0.0001 | 0.5689 | 0.4338 |
| | Postyield Work (mJ) | 10.14 ± 4.01 | 3.99 ± 1.46 | 8.62 ± 2.55 | 2.73 ± 0.85 | 0.0507 | <0.0001 | 0.8493 |
| | Total Work (mJ) | 11.15 ± 4.23 | 4.98 ± 1.46 | 9.22 ± 2.66 | 3.43 ± 0.84 | 0.0206 | <0.0001 | 0.797 |
| | Yield Stress (MPa) | 93.8 ± 18.9 | 108.9 ± 19.1 | 87.9 ± 16.4 | 110.9 ± 15.9 | 0.6816 | 0.0002 | 0.4071 |
| | Ultimate Stress (MPa) | 123 ± 9 | 137 ± 13 | 120 ± 10 | 127 ± 16 | 0.0489 | 0.002 | 0.4009 |
| | Strain to Yield (mε) | 20.7 ± 4.6 | 20.8 ± 3.2 | 17.3 ± 2.8 | 18.9 ± 2.8 | 0.0057 | 0.373 | 0.4084 |
| | Total Strain (mε) | 160.4 ± 83 | 71.6 ± 30.5 | 178.9 ± 64.7 | 69.9 ± 21.7 | 0.5885 | <0.0001 | 0.5153 |
| | Modulus (GPa) | 5.09 ± 0.61 | 5.85 ± 0.87 | 5.82 ± 0.70 | 6.66 ± 0.92 | 0.0005 | 0.0003 | 0.8355 |
| | Resilience (MPa) | 1.10 ± 0.44 | 1.25 ± 0.34 | 0.87 ± 0.25 | 1.14 ± 0.27 | 0.0686 | 0.0263 | 0.4908 |
| | Toughness (MPa) | 12.17 ± 4.57 | 6.40 ± 1.93 | 13.38 ± 3.50 | 5.56 ± 1.28 | 0.8287 | <0.0001 | 0.2336 |
| | | Male | | Female | | P-value | | |
| | | WT (n = 15) | G610C (n = 15) | WT (n = 15) | G610C (n = 15) | Sex | Genotype | Sex * Genotype |
| **16 WKS** | Yield Force (N) | 13.3 ± 3.0 | 13.9 ± 3.5 | 10.7 ± 1.6 | 11.54 ± 1.8 | 0.0005 | 0.2593 | 0.8558 |
| | Ultimate Force (N) | 17.7 ± 2.4 | 16.7 ± 3.6 | 16.5 ± 1.4 | 13.9 ± 2.3 | 0.0037 | 0.0083 | 0.2216 |
| | Displacement to Yield (µm) | 117 ± 14 | 141 ± 22 | 101 ± 18 | 126 ± 12 | 0.0006 | <0.0001 | 0.9092 |
| | Postyield Displacement (µm) | 598 ± 330 | 206 ± 108 | 818 ± 281 | 264 ± 108 | 0.0224 | <0.0001 | 0.1764 |
| | Total Displacement (µm) | 714.9 ± 324 | 347.5 ± 98 | 919 ± 288 | 390.3 ± 109 | 0.0412 | <0.0001 | 0.178 |
| | Stiffness (N/mm) | 127 ± 15 | 109 ± 22 | 123 ± 13 | 104 ± 18 | 0.3457 | <0.0001 | 0.9114 |
| | Work to Yield (mJ) | 0.88 ± 0.28 | 1.08 ± 0.38 | 0.62 ± 0.19 | 0.81 ± 0.17 | 0.0002 | 0.0061 | 0.9146 |
| | Postyield Work (mJ) | 7.41 ± 2.65 | 2.67 ± 0.92 | 8.99 ± 2.4 | 2.84 ± 0.87 | 0.0782 | <0.0001 | 0.1524 |
| | Total Work (mJ) | 8.29 ± 2.66 | 3.76 ± 0.83 | 9.61 ± 2.46 | 3.65 ± 0.9 | 0.2232 | <0.0001 | 0.1521 |
| | Yield Stress (MPa) | 100.3 ± 13.8 | 125.6 ± 20.6 | 88.3 ± 11.8 | 121.8 ± 15.3 | 0.0566 | <0.0001 | 0.3209 |
| | Ultimate Stress (MPa) | 135 ± 7 | 152 ± 15 | 137 ± 8 | 147 ± 17 | 0.6723 | 0.0002 | 0.2776 |
| | Strain to Yield (mε) | 17.8 ± 2.4 | 20.9 ± 3.7 | 15.5 ± 2.8 | 17.7 ± 2.1 | 0.0003 | 0.0005 | 0.4824 |
| | Total Strain (mε) | 107.8 ± 48.2 | 51.3 ± 14.1 | 140.3 ± 42.4 | 54.6 ± 15.4 | 0.0446 | <0.0001 | 0.0996 |
| | Modulus (GPa) | 6.45 ± 0.67 | 6.73 ± 1.12 | 6.70 ± 0.76 | 7.88 ± 1.29 | 0.0081 | 0.0061 | 0.0841 |
| | Resilience (MPa) | 1.00 ± 0.24 | 1.44 ± 0.42 | 0.79 ± 0.23 | 1.19 ± 0.20 | 0.0022 | <0.0001 | 0.7913 |
| | Toughness (MPa) | 9.58 ± 2.81 | 5.16 ± 1.20 * | 12.27 ± 3.23 $ | 5.38 ± 1.28 * | 0.0184 | <0.0001 | 0.044 |

Significant effects (p>0.05) are highlighted in gray. Significant post-hoc values (p<0.05) between genotypes are marked with *, and between sex are marked with $. Data presented as mean +/- standard deviation.

have been successfully performed with minimal specimen loss [33]. However, these mice can be difficult to acquire as they are not commercially available. Although additional models exist (e.g. CRTAP [34], Swaying [35], Jrt [36], Col1a1$^{\pm365}$[37]), the G610C model may be preferable as it is commercially available, is reflective of a mutation found in humans, and is severe enough to be relevant but moderate enough to be practical.

Results from the current study show that G610C mice bred on a B6 background have small, highly mineralized bones with lower structural strength and decreased deformability. Tissue-level analyses demonstrate that this skeletal weakness is not only attributable to the smaller body and bone sizes of the diseased mice, but also to inferior tissue quality. G610C bones

exhibited brittle behavior, fracturing at lower displacements and strains than their WT counterparts (Figs 3 and 4), despite showing largely similar tissue-level strength versus WT bones (as measured by ultimate stress). This finding was true for both male and female mice, and at ages reflecting adolescence through skeletal maturity. Brittle behavior is expected of an OI phenotype due to a lack of normal collagen and subsequent aberrant mineralization which can manifest as increased or decreased tissue mineral density, demonstrating that the observed structural weakness is caused, in part, by inferior tissue quality in addition to reduced bone quantity. G610C tissue consistently showed higher mineralization than WT tissue (as measured by TMD which is reflective of the mineralization of the tissue itself without the confounding factor of bone mass), a trend which has been seen before in G610C [6, 12], as well as oim mice [38]. Importantly, no spontaneous in vivo fractures were observed in this study. Together, these findings demonstrate that this model is a faithful representation of mild to moderate OI and a solid test bed for intervention studies aimed at mitigating defects in quantity and quality.

The results presented here agree with and expand upon previously published data of the G610C mechanical phenotype, despite the use of diverse methodologies. Jeong *et al.* performed torsion tests on femora from 16 wk male and female femora and saw the same general trend of G610C bones having lower ultimate strength and performing lower total work than WT [25]. Mertz *et al.* performed 4-pt bending tests on 16 wk male femora and saw a significant decrease in total work, post-yield work, and post-yield displacement, and significant increase in displacement to yield [17]. Omosule *et al.* and Bi *et al.* both tested 16 wk femora from female mice in 3-pt bending, and saw a significant decrease in ultimate force, post-yield work, total work, and post-yield displacement [21, 23], as well as ultimate stress and modulus [21]. Omosule *et al.* also analyzed femora from male mice and saw higher ultimate force values in G610C than in WT, however this is inconsistent with our results and other published data. Bateman *et al.* tested femora and tibiae from 8 wk male mice in 4-pt bending, and saw similar results to the 10 wk data reported here, with ultimate force, work to yield, post-yield work, and stiffness being or trending significantly lower in G610C mice in both bones, with the exception of a reverse trend for tibial work-to-yield (inconsistent with our results) [18]. Lee *et al.* performed 4-pt bending with tibiae from 15 wk female mice, and also reported a significant decrease in ultimate force and work to yield [20]. Finally, Little *et al.* performed 4-pt bending on tibiae from 9 wk female mice and saw a significant decrease in ultimate force and post-yield work [19]. In summary, ultimate force and post-yield work are consistently lower in G610C mice than in WT, regardless of age or sex. Stiffness and work to yield either do not change between genotypes or are slightly reduced in G610C, with femora and tibiae behaving slightly differently. Importantly, no study to date has looked at the mechanical behavior of G610C in both hindlimb bones, both sexes, or two different age points.

Mechanical studies of G610C have been performed with mice bred on a variety of background strains other than C57BL/6, but since phenotype severity is strongly dependent on background strain, as shown by Daley *et al.* [6], results from these studies cannot be directly compared with the data presented here. As has been briefly summarized, previous studies with mice bred on a B6 background have only reported basic structural mechanical properties (with the exception of Bi *et al.* [21]), and are often statistically limited by smaller sample sizes (e.g. n = 5). By estimating tissue-level defects in addition to structural deficiencies and increasing statistical power by using larger groups, the characterization reported here provides an expanded analysis of the underlying causes of skeletal weakness and brittleness in the G610C model of OI.

This study is limited by its scope, which focused on the architecture and mechanics of commonly analyzed long bones (hindlimbs) at two important developmental age points. It is not

known how the phenotype might evolve as these mice continue to age, nor if there are similar phenotypes in other skeletal and non-skeletal locations. The impacts of cellular processes were also not examined, nor was consideration made for other measures of tissue quality, including characterization of tissue composition (beyond TMD) or collagen structure. These experiments were beyond the scope of the current investigation but will be considered in future work.

In conclusion, the Amish *Col1a2*[G610C/+] model of OI exhibits a distinctly weak, brittle phenotype that similar in both males and female mice, and at two distinct and important developmental ages. Importantly, no spontaneous fractures were noted in any bones harvested in the current study. This characterization demonstrates that despite being a moderate OI model, the Amish G610C mouse model bred on a B6 background strain maintains a distinctly brittle phenotype and is well-suited for use in future intervention studies.

## Supporting information

**S1 Table. LV4 trabecular properties at 10 and 16 weeks, with p-values from 2-way ANOVA.** Significant effects (p>0.05) are highlighted in gray. Significant post-hoc values (p>0.05) between genotypes are marked with *, and between sex are marked with $. Data presented as mean +/- standard deviation.
(DOCX)

**S1 Fig. Cortical cross sections from tibiae.** Shown are average profiles of the tibia cortical ROIs from each group, clearly demonstrating that cortical bone area is consistently smaller in G610C mice in both ages and sexes.
(TIF)

**S2 Fig. Tibial and femoral cortical bone area and total area for all groups.** Data shown is mean + standard deviation. Main effects (p-value < 0.05 from 2-way ANOVA) of sex (#) and genotype (*) were seen for all groups, with no interaction.
(TIF)

**S3 Fig. Bone volume fraction of trabecular, femoral, and vertebral bone for all groups.** Data shown is mean + standard deviation. Main effects (p-value < 0.05 from 2-way ANOVA) of sex (#) and genotype (*) were seen for all groups. There was a significant interaction effect in only vertebral data; p-values shown are from Tukey post-hoc tests.
(TIF)

**S4 Fig. Sagittal femoral cross-sections.** Shown are cross-sectional views of 10-wk femurs from representative mice (the trend is similar in 16-wk mice). Trabecular bone quantity varies significantly between genotypes.
(TIF)

**S5 Fig. Coronal vertebral cross-sections.** Shown are cross-sectional views of 10-wk L4 vertebrae from representative mice (the trend is similar in 16-wk mice). Trabecular bone quantity varies significantly between genotypes.
(TIF)

## Author Contributions

**Conceptualization:** Joseph M. Wallace.

**Data curation:** Rachel Kohler.

**Formal analysis:** Rachel Kohler, Carli A. Tastad, Joseph M. Wallace.

**Funding acquisition:** Joseph M. Wallace.

**Investigation:** Rachel Kohler, Carli A. Tastad.

**Methodology:** Joseph M. Wallace.

**Project administration:** Joseph M. Wallace.

**Resources:** Amy Creecy.

**Supervision:** Amy Creecy, Joseph M. Wallace.

**Validation:** Carli A. Tastad.

**Visualization:** Rachel Kohler.

**Writing – original draft:** Rachel Kohler, Carli A. Tastad.

**Writing – review & editing:** Rachel Kohler, Joseph M. Wallace.

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
