## [Decision Letter · Decision Letter 0]

4 Jun 2021

PONE-D-21-12105

Morphological and Mechanical Characterization of Bone Phenotypes in the Amish G610C Murine Model of Osteogenesis Imperfecta

PLOS ONE

Dear Dr. Wallace,

Thank you for submitting your manuscript to PLOS ONE. After careful consideration, we feel that it has merit but does not fully meet PLOS ONE’s publication criteria as it currently stands. Therefore, we invite you to submit a revised version of the manuscript that addresses the points raised during the review process.

Both reviewers were generally positive but offered specific suggestions related to a more thorough discussion of results in light of other studies/models, additional measurements, and the presentation of methods and results.

We look forward to receiving your revised manuscript.

Kind regards,

Ryan K. Roeder, PhD

Academic Editor

PLOS ONE

Journal Requirements:

Reviewers' comments:

Reviewer's Responses to Questions

**Comments to the Author**

1. Is the manuscript technically sound, and do the data support the conclusions?

Reviewer #1: Yes

Reviewer #2: Yes

2. Has the statistical analysis been performed appropriately and rigorously? 

Reviewer #1: Yes

Reviewer #2: Yes

3. Have the authors made all data underlying the findings in their manuscript fully available?

Reviewer #1: Yes

Reviewer #2: Yes

4. Is the manuscript presented in an intelligible fashion and written in standard English?

Reviewer #1: Yes

Reviewer #2: Yes

5. Review Comments to the Author

Reviewer #1: This is a clearly written and thorough study on the mechanical properties of the G610C OI model. The authors are to be commended on an excellent and important study.

I strongly suggest, though leave to the authors to decide whether to defer this for future study, if necessary, that the study also include an assessment of the collagen content and architecture. Since this phenotype is driven by collagen mutation, the collagen content and organization are essential to understand the tissue mechanics. Ash weight/dry weight ratios, polarized picrosirius red microscopy, SHG imaging, hydroxyproline content, and/or other assays of the matrix collagen would help place these helpful findings in context and provide mechanistic insight into how this mutation alters bone mechanics.

Reviewer #2: This study describes bone morphological and mechanical characterization of the Amish

Col1a2G610C/+ (G610C) mouse model of mild to moderate osteogenesis imperfecta (OI). As this model is currently being used in a range of intervention studies, it’s important to have baseline characterization of the phenotype, in particular in both males and females. Thus, these data provide important information to the research community seeking to use these mice.

The mice were analyzed at 10 and 16 weeks, defined as “crucial developmental ages”. It would be helpful to understand why those specific timepoints were chosen, and if the possibility exists of adding older mice to the study. Nevertheless, the data are interesting. However, the Discussion and comparison to data from other studies is considerably lacking. The authors should add in specific comparison to data previously collected from these mice at similar ages, in spite of the fact that they aren’t exactly the same mice. Further, the authors appear to try to be dissuading researchers from using the oim/oim mouse model, due to it’s spontaneous fractures, and potential difficulty in breeding. There is clearly space in the realm of mouse models of OI for both models, and the authors should highlight why one would be used compared to another, instead of a primarily negative view of the widely used oim/oim mice. Is it possible that some studies would benefit from a therapeutic intervention in a mouse model of OI that fractured spontaneously, while others would benefit from a less severe model, such as the G610C? It’s important to add these aspects into the Discussion to make it more balanced. Further, comparisons to the material properties of the BRTL and oim/oim mice that have been previously established should be discussed. Together, these 3 models provide a range of opportunities for investigating therapeutics for this devasting disease, and information presented here adds to the ability to go forward with this type of research.

Specific Comments:

1. Introduction: please add a reference for this statement related to the OIM mouse model that is commonly used: “…while this substitution genotypically models Ehlers-Danlos syndrome in humans”. Is that correct?

2. Methods: please add inan explanation for 3 pt bend tests in femurs and 4 pt bend tests in tibia, and for the orientation of the bones being different in the two testing approaches.

3. Results: It would be helpful to have supplemental data that shows bar graphs for all the parameters. The Tables are very challenging to digest.

4. Figure 1 legend is not correct.

5. Overall, the figure quality in the pdf is not good.

Discussion:

6. Please add in reference(s) for this sentence on page 12: OI is characterized by bones that are not only small, but also suffer from inferior tissue quality due to mutant collagen being incorporated into the bone matrix thereby inhibiting normal crosslinking, mineralization, and hydration.

7. p 13, “In addition, due to their poor health, breeding these (oim) mice is challenging and those that survive often die prior to study endpoints”…. “making it difficult to power interventions studies”. I don’t believe these statements are accurate. There are multiple studies in the literature that use these mice. They are the most widely used mouse model of OI that fracture spontaneously, it seems disingenuous to discourage use of them.

8. Page 14, this sentence should be modified to include the fact the phenotype is of mild OI , eg, no fractures occurring. “these findings demonstrate that this model is a faithful representation of (mild -to-moderate) OI and a solid test bed for intervention studies aimed at mitigating defects in quantity and quality.

9. A more detailed comparison to other mouse models of OI should be added in. Reduced quality of matrix has been demonstrated in several other models as well, and should be mentioned here.

10. Add in discussion of why tissue density is higher in the G610C mice. Has this been found in other OI mouse models?

6. PLOS authors have the option to publish the peer review history of their article (what does this mean?). If published, this will include your full peer review and any attached files.

Reviewer #1: No

Reviewer #2: No

---

## [Author Response · Author response to Decision Letter 0]

12 Jul 2021

Response to Reviewers

Reviewer 1: I strongly suggest, though leave to the authors to decide whether to defer this for future study, if necessary, that the study also include an assessment of the collagen content and architecture. Since this phenotype is driven by collagen mutation, the collagen content and organization are essential to understand the tissue mechanics. Ash weight/dry weight ratios, polarized picrosirius red microscopy, SHG imaging, hydroxyproline content, and/or other assays of the matrix collagen would help place these helpful findings in context and provide mechanistic insight into how this mutation alters bone mechanics.

A: We appreciate the points being made by the reviewer. Understanding multiscale mechanisms driving a phenotype such as one in an OI model is critical. However, these types of assays are complex and labor intensive. We do plan to assess and characterize collagen in future studies using this model, including treatment studies, but those analyses were beyond the scope of this project which was focused on an overall initial bone phenotype characterization.

Reviewer 2: The mice were analyzed at 10 and 16 weeks, defined as “crucial developmental ages”. It would be helpful to understand why those specific timepoints were chosen, and if the possibility exists of adding older mice to the study. 

A: These timepoints were chosen for two reasons. First, as stated, they bookmark an important stage in mouse skeletal development, being the transition from adolescence (10 wks) to mature adulthood (16 wks). This correlates with the period of adolescent skeletal growth in human OI, which is the most critical period for intervention. Second, for that reason, these are the ages most commonly used in loading and intervention studies of OI (as can be seen from the literature summary), and therefore are the most useful to characterize. While older mice could be added to the study, they would not be as clinically or experimentally relevant, as, again, interventions are most needed for pediatric and adolescent patients. As we move forward with intervention studies using mice at 10-16 weeks of age, our plan is to complete additional characterization studies in older OI mice. 

R2: The Discussion and comparison to data from other studies is considerably lacking. The authors should add in specific comparison to data previously collected from these mice at similar ages, in spite of the fact that they aren’t exactly the same mice.

A: Thank you for pointing out this limitation of the current manuscript. Specific comparisons from other studies have now been added to the discussion to address this concern.

R2: The authors appear to try to be dissuading researchers from using the oim/oim mouse model, due to it’s spontaneous fractures, and potential difficulty in breeding. There is clearly space in the realm of mouse models of OI for both models, and the authors should highlight why one would be used in compared to another instead of a primarily negative view of the widely used oim/oim mice. Is it possible that some studies would benefit from a therapeutic intervention in a mouse model of OI that fractured spontaneously, while others would benefit from a less severe model, such as the G610C? It’s important to add these aspects into the Discussion to make it more balanced.

A: Perhaps our wording was too firm since we absolutely agree that the oim/oim model is useful and appropriate for investigating many research questions. We have worked with the oim/oim model for years and have multiple relevant publications. However, we are arguing that it is not ideal for the specific use cases of mechanical intervention and bending analysis. In vivo loading with oim/oim mice is more likely to cause fracture. In addition, because of sample loss due to spontaneous fracture and the need for large sample sizes to overcome variability in mechanical analyses, we believe that studies that use loading as an intervention and/or intend to perform mechanical analysis as a primary study endpoint would be better served with less severe models, such as the G610C model. We have edited some of our wording in the discussion to clarify that the concern is with a certain type of study, and not the oim/oim model overall. However, as a review of the oim/oim model is not the focus of this paper, we do not think it is necessary to list alternative uses the model can have, but instead leave that investigation to the reader.

R2: Comparisons to the material properties of the BRTL and oim/oim mice that have been previously established should be discussed. Together, these 3 models provide a range of opportunities for investigating therapeutics for this devastating disease, and information presented here adds to the ability to go forward with this type of research.

A: The overall phenotype of BRTL and oim/oim mice was briefly covered in the discussion. As the focus of the paper is characterizing the G610C model, a thorough discussion of the material properties of other models is outside the scope of this paper.

R2: Introduction- please add a reference for this statement related to the OIM moue model that is commonly used. “… while this substitution genotypically models Ehlers-Danlos syndrome in humans”. Is that correct?

A: Ehlers-Danlos syndrome is a group of a genetic connective tissue disorders, many of which are caused by splice mutations in the COL1A2 gene that inhibit synthesis of Type 1 alpha 2 chains, causing collagen fibrils to be composed of alpha 1 homotrimers. In humans, this mutation can occur without the presentation of OI (although depending on the exon site a patient might have both), while in mice this same mutation presents as OI. Two relevant citations have been added for the reader’s reference.

R2: Methods- please add in an explanation for 3 pt bend tests in femurs and 4 pt bend tests in tibia, and for the orientation of the bones being different in the two testing approaches.

A: Different tests were used for femurs and tibia due to the differences in bone size and shape, and this choice is based on the experience of 20 years of mechanically testing mouse bones. Generally, 4-pt bending is the preferred bending method as the region of the bone between the loading points is under pure bending without contribution from shear stress. However, the shorter the bone being tested, the closer the two loading points must be placed together, and the difference between the two test types becomes less significant. In addition, for a given force and loading span, the bending moment is larger in 3-pt versus 4-pt bending meaning larger loads are necessary to cause failure in 4-pt bending. For a robust bone with a thick cortex like a tibia, these higher loads are fine and 4-pt bending is preferred. However, the thinner femur can undergo ring-type deformation at higher loads causing the cross section to change shape (which violates a main assumption of the test). The deformation in the cross section also overestimates the overall deflection of the bone, making it appear less stiff than it is. As can be seen from the cited literature, both loading types are used with femurs, with near-identical results. Since either method is usable, we chose to use 3-pt bending for femurs to reduce error and increase consistency.

Femora and tibiae are loaded in different orientations based on shape. To get consistent and accurate mechanical data, the specimen must not move during the test, so bone orientation is chosen strategically to minimize the likelihood of rolling by increasing stability. In addition, one main assumption of bending tests is a constant prismatic cross section. This is easily accomplished with the femur since it is more or less elliptical in shape. The tibia changes shape over the region being tested but the moment of inertia about the anterior-posterior axis (in the medial-lateral direction) is more or less constant over that region. Taking all of this into account, femurs are loaded with the posterior surface in tension (so that the longer axis of the elliptical cross-section is perpendicular to the load), and tibiae are loaded with the medial surface in tension. 

R2: Results - It would be helpful to have supplemental data that shows bar graphs for all the parameters. The Tables are very challenging to digest.

A: We understand that tables are difficult to interpret for the major trends of the data. However, because this is a characterization paper, we believe they are the most complete method of reporting the data and have attempted to add some color coding to emphasize differences between groups. As an aid to the reader, we have added two figures to the supplemental information that show bar graphs of cortical data (in bone area and total area) and trabecular data (percent bone volume).

R2: Figure 1 legend is not correct, overall the figure quality in the pdf is not good.

A: Figure 1 legend has been checked and is accurate. The figure quality shown in the of the pdf is not representative of the figure quality submitted.

R2: Please add in reference(s) for this sentence on page 12: OI is characterized by bones that are not only small, but also suffer from inferior tissue quality due to mutant collagen being incorporated in the bone matrix thereby inhibiting normal crosslinking, mineralization, and hydration.

A: While the main point of this sentence is essentially a definition of the OI phenotype, to bolster its validity a reference has been added to a source that specifically looked at the multiscale effects of mutant collagen in murine OI bone.

R2: p 13, “in addition, due to their poor health, breeding these (oim) mice is challenging and those that survive often die prior to study endpoints”… “making it difficult to power intervention studies”. I don’t believe these statement are accurate. There are multiple studies in the literature that use these mice. They are the most widely used mouse model of OI that fracture spontaneously, it seems disingenuous to discourage use of them.

A: As discussed previously, we did not intend to discourage the broad use of the oim/oim model. While we believe the point remains that there are higher rates of animal loss in severe models, which could be an important factor to consider when choosing the appropriate model for a study, we have removed this sentence to be less negative toward the oim model.

R2: p 14, this sentence should be modified to include the fact that the phenotype is of mild OI, eg, no fractures occurring. “these findings demonstrate that this model is a faithful representation of (mild to moderate) OI and a solid test bed for intervention studies aimed at mitigating defects in quantity and quality.

A: The suggested edit has been added.

R2: A more detailed comparison to other mouse models of OI should be added in. Reduced quality of matrix has been demonstrated in several other models as well, and should be mentioned here.

A: Yes, any OI model would be expected to have reduced matrix properties and this has indeed been found in other mouse models. However, the focus of this paper was not an analysis of OI mouse models but a characterization of specifically the G610C Amish model. A more detailed comparison to other models is beyond the scope of this paper but references to other models are included.

R2: Add in discussion of why tissue density is higher in G610C mice Has this been found in other OI mouse models”

A: Tissue mineral density, or increased mineralization, is an example of the “aberrant mineralization” briefly mentioned in the discussion, caused by defective collagen, and is believed to be one of the causes of increased brittleness and reduced toughness in the OI model. Increased TMD has been seen previously in the G610C and other OI models. References and a short expansion of this topic have been added to the discussion.

---

## [Editor Report · Decision Letter 1]

14 Jul 2021

Morphological and Mechanical Characterization of Bone Phenotypes in the Amish G610C Murine Model of Osteogenesis Imperfecta

PONE-D-21-12105R1

Dear Dr. Wallace,

We’re pleased to inform you that your manuscript has been judged scientifically suitable for publication and will be formally accepted for publication once it meets all outstanding technical requirements.

Kind regards,

Ryan K. Roeder, PhD

Academic Editor

PLOS ONE
---

## [Editor Report · Acceptance letter]

19 Aug 2021

PONE-D-21-12105R1 

Morphological and Mechanical Characterization of Bone Phenotypes in the Amish G610C Murine Model of Osteogenesis Imperfecta 

Dear Dr. Wallace:

I'm pleased to inform you that your manuscript has been deemed suitable for publication in PLOS ONE. Congratulations! Your manuscript is now with our production department. 

Kind regards, 

on behalf of

Dr. Ryan K. Roeder 

Academic Editor

PLOS ONE